# Parkinson’s Disease Through the Lens of Metabolomics: A Targeted Systematic Review on Human Studies (2019–2024)

**DOI:** 10.3390/jcm14176277

**Published:** 2025-09-05

**Authors:** Federico Cannas, Karolina Krystyna Kopeć, Natalia Zuddas, Flaminia Cesare Marincola, Giorgio Arcara, Michele Loi, Michele Mussap, Vassilios Fanos

**Affiliations:** 1Department of Mechanical, Chemical and Materials Engineering, University of Cagliari, 09123 Cagliari, Italy; fcannas1@gmail.com (F.C.); k.kopec@studenti.unica.it (K.K.K.); 2Department of Chemical and Geological Sciences, University of Cagliari, Monserrato, 09042 Cagliari, Italy; nataliazuddas02@gmail.com; 3IRCCS San Camillo Hospital, 30121 Venice, Italy; giorgio.arcara@hsancamillo.it; 4Neonatal Intensive Care Unit, Department of Surgical Sciences, University Hospital of Cagliari, Monserrato, 09042 Cagliari, Italy; micheleloi1994@gmail.com (M.L.); mmumike153@gmail.com (M.M.); vafanos@tiscali.it (V.F.); 5Laboratory Unit, Department of Surgical Sciences, University Hospital of Cagliari, Monserrato, 09042 Cagliari, Italy

**Keywords:** Parkinson’s disease, metabolomics, biomarker discovery, early diagnosis

## Abstract

**Background:** Parkinson’s disease (PD) is a chronic, progressive neurodegenerative disorder characterized by motor and non-motor symptoms. As conventional diagnostic methods are limited in their ability to detect early-stage PD or monitor its progression, there is growing interest in identifying molecular biomarkers with clinical utility. This systematic review synthesizes recent advancements in the application of metabolomics to PD, with a specific focus on human studies published between 2019 and 2024, a period of notable growth in the research area. **Methods:** Following PRISMA 2020 guidelines, a comprehensive literature search was conducted across major scientific databases. After screening, 16 eligible original studies were selected based on predefined criteria. Key features extracted included study design, biofluid type, analytical platform, statistical approach, and main findings. **Results:** Consistent metabolic alterations were observed across several biological pathways, including amino acid metabolism, lipid regulation, mitochondrial energy production, oxidative stress, polyamine metabolism, as well as in gut microbiota-derived metabolites. Biofluids analyzed included plasma, serum, cerebrospinal fluid, saliva, urine, and sebum. While plasma and serum remained the most studied matrices, emerging interest in non-invasive fluids such as saliva and sebum reflects their potential in clinical settings. Methodological heterogeneity was noted across studies, particularly in confounder adjustment and study design. **Conclusions:** Despite certain limitations, the included studies collectively point to the potential of metabolomics in identifying robust diagnostic and prognostic signatures for PD. This review emphasizes the need for longitudinal studies, methodological standardization, and integration with other omics approaches to advance biomarker discovery and support the development of precision medicine strategies for PD.

## 1. Introduction

Parkinson’s disease (PD) is a chronic and progressive neurodegenerative disorder that primarily affects the central nervous system [1]. It is marked by the gradual degeneration of dopaminergic neurons in the substantia nigra, resulting in a dopamine deficiency that disrupts motor control and coordination [2,3,4,5]. As a result, patients develop a range of motor symptoms, including tremors, rigidity, bradykinesia (slowness of movement), and postural instability that significantly impair the patient’s quality of life and autonomy. Beyond these motor impairments, PD is also associated with non-motor symptoms such as cognitive decline, mood disorders, altered sleep patterns, autonomic dysfunction, and gastrointestinal issues [6,7,8]. As the disease progresses, the ability to cope with these daily challenges increases, leading to functional limitations and growing disability. Patients face increasing functional limitations, higher fall risk, swallowing difficulties, and vulnerability to comorbidities [9,10]. Given the complex and multifaceted nature of PD, effective management requires a holistic and multidisciplinary approach, incorporating the expertise of neurologists, rehabilitation specialists, psychiatrists, and social workers to address the diverse and evolving needs of patients and enhance their quality of life [11].

PD is the second most common neurodegenerative disorder after Alzheimer’s disease, with a global prevalence of around 1–2% in people over 65 years old. Over the past quarter-century, the incidence of PD has doubled, with over 8.5 million individuals worldwide affected as of 2019 [11]. This growing prevalence also entails considerable economic implications, both at the individual and healthcare system levels. The direct costs of PD, including medication, hospital stays, rehabilitation services, and long-term care, rise steadily as the disease advances. Indirect costs, such as reduced productivity due to disability and informal care provided by family members, further exacerbate financial strain. These rising costs pose a growing challenge to healthcare systems worldwide, particularly given the chronic and progressive nature of the disease. Thus, managing the economic impact of PD requires long-term strategic investments in both medical and social care [11].

Despite significant research efforts, the pathophysiology of PD remains only partially elucidated. Its onset and progression are thought to result from a complex interplay of genetic predisposition, environmental exposures, and neurochemical alterations. Several mechanisms have been implicated in the disease process, including the accumulation of misfolded protein aggregates, particularly α-synuclein, mitochondrial dysfunction, oxidative stress, and neuroinflammation [2,12]. Emerging evidence also points to the gut–brain axis as a contributor to PD pathogenesis [13]. Disruptions in microbial composition and activity have been linked to increased intestinal permeability and systemic inflammation, which may subsequently trigger microglial activation. Activated microglia release pro-inflammatory cytokines and reactive oxygen species, accelerating dopaminergic neuron loss. Given their heterogeneity and dynamic shifts during aging and neurodegeneration, microglia are increasingly viewed as potential biomarkers and therapeutic targets, alone or alongside other disease-modifying strategies [14].

Although the mechanism by which mutations in different regions of protein kinase PARK6/PINK1 lead to early-onset forms of Parkinson’s disease has been recently elucidated [15], the precise cascade of events leading to dopaminergic neuron degeneration and disease progression remains only partially understood. This incomplete knowledge hinders the early recognition of the disease and, in turn, the development of targeted therapies capable of halting or slowing disease progression. This limitation is further exacerbated by the heterogeneity in PD symptoms, which give rise to distinct clinical phenotypes [16,17] often overlapped with other neurodegenerative disorders, such as Progressive Supranuclear Palsy (PSP) [18]. This overlap complicates differential PD diagnosis particularly in the early stages, as these conditions may share similar motor features but they differ significantly in their pathophysiology, progression, and response to treatment. As a result, to date, no single diagnostic test exists for PD. Diagnosis is primarily based on clinical evaluation, often complemented by advanced neuroimaging techniques such as magnetic resonance imaging (MRI) and positron emission tomography (PET) [8,19,20].

## 2. The Potential Role of Metabolomics for Parkinson’s Disease

The diagnostic complexity of PD highlights the urgent need for reliable biomarkers capable of detecting the disease at its earliest stages, distinguishing it from related neurological conditions, and tracking its progression over time. In this regard, metabolomics offers promising opportunities for identifying novel, non-invasive biomarkers that reflect underlying biochemical alterations in PD, with potential implications for diagnosis, prognosis, and therapeutic development [21,22,23,24]. Metabolomics refers to the comprehensive profiling of the metabolome, the complete set of low-molecular-weight (<1.5 kDa) metabolites produced by an organism at a given time [25]. As the endpoint of genomic, transcriptomic, and proteomic processes, the metabolome provides a dynamic snapshot of the organism’s physiological or pathological status. Advancements in analytical technologies, particularly nuclear magnetic resonance (NMR) spectroscopy and mass spectrometry (MS)-based platforms, have significantly expanded the metabolomic landscape of PD. These studies have revealed consistent metabolic alterations in lipid metabolism, energy pathways, amino acid turnover, and gut microbiome-derived metabolites, providing new insights into disease mechanisms and potential biomarkers. Notably, these metabolic signatures are often stage-specific and influenced by multiple factors such as genetic mutations (e.g., LRRK2, GBA), age, sex, comorbidities, and environmental exposures [22,24,26,27,28,29,30,31,32,33]. Despite this complexity, metabolomics has proven to be a powerful approach not only for deepening the understanding of PD pathophysiology but also paving the way toward novel diagnostic and prognostic biomarkers that can support precision medicine strategies.

Several recent reviews have highlighted the contribution of metabolomics in PD, emphasizing its capacity to uncover systemic and CNS-specific biochemical alterations. Li et al. analyzed twenty clinical metabolomics studies and identified recurrent alterations in amino acid and lipid metabolism [34]. They also highlighted the lack of longitudinal studies and underscored the importance of biomarker validation across independent cohorts, emphasizing how pharmacological treatments (e.g., L-DOPA) may significantly influence metabolite levels. Ciobanu et al. expanded the focus by including non-motor psychiatric symptoms, such as depression, anxiety, and psychosis, and their metabolic correlates across various biofluids (e.g., 5-HIAA, kynurenines, 8-OhdG, α-synuclein), underscoring the link between peripheral metabolism, neuroinflammation, and psychiatric burden in PD [35]. Troisi et al. offered a methodological and conceptual perspective by framing metabolomics within a systems biology approach [36]. They mapped metabolite changes onto major PD mechanisms such as mitochondrial dysfunction, oxidative stress, lipid dysregulation, and α-synuclein aggregation and emphasized how both targeted and untargeted metabolomics across plasma, CSF, and brain tissue can reveal converging signatures (e.g., citrate, sphingolipids, 3-hydroxykynurenine), influenced by sex, disease stage, and sample handling. One of the most comprehensive efforts to date is the meta-analysis by Luo et al. which integrated data from 74 studies to identify 190 reproducible alterations in metabolite levels [37]. This work reconstructed a PD-specific metabolic network, highlighting disrupted pathways including those involved in dopamine and tyrosine metabolism, the kynurenine pathway, the urea cycle, and caffeine metabolism.

Building on these foundational studies, the present review offers a timely and systematic synthesis of clinical metabolomics research published between 2019 and 2024. Different from earlier reviews, which often included smaller or heterogeneous cohorts, this review applies stricter inclusion criteria, such as a minimum sample size of 50 participants and the exclusive inclusion of studies conducted on human subjects. Furthermore, the current review provides a comparative analysis of underutilized versus traditionally used biofluids, critically assesses study quality using the Newcastle–Ottawa Scale, and discusses methodological limitations and statistical strategies. This comprehensive and quality-focused approach ensures that the review not only summarizes recent findings but also serves as a reference framework for improving the design and clinical translatability of future metabolomics research in PD.

## 3. Methods

The selection algorithm used in this systematic review followed the Preferred Reporting Items for Systematic Reviews and Meta-Analyses (PRISMA) guidelines [38] to ensure a systematic and reproducible process (Figure 1). This systematic review was not prospectively registered in any public database.

### 3.1. Search Strategies and Selection Criteria

To conduct a comprehensive literature search, three prominent scientific databases were utilized: ISI Web of Science (WOS), Elsevier Science Direct, and NIH PubMed. These platforms were selected for their extensive repositories of peer-reviewed journals, which ensure wide coverage and access to high-quality, relevant studies. The selection of papers for this review followed specific criteria to ensure the inclusion of relevant and up-to-date research on the role of metabolomics in PD:Keywords: The databases were queried to look for specific keywords in their title, such as “Parkinson’s disease”, “NMR spectroscopy”, “mass spectrometry”, “metabolomics”. The exact search strings used for three databases were as follows:
NIH PubMed query: (“Parkinson’s disease”[Title]) AND (“metabolomics”[Title] OR “NMR spectroscopy”[Title] OR “mass spectrometry”[Title] OR “metabolites”[Title]);Science Direct query: (“Parkinson’s disease”[Title]) AND (“metabolomics”[Title] OR “NMR spectroscopy”[Title] OR “mass spectrometry”[Title] OR “metabolites”[Title]);WOS query: TI = (“Parkinson’s disease”) AND TI = (“metabolomics” OR “NMR spectroscopy” OR “mass spectrometry” OR “metabolites”).
Publication date: To ensure the relevance and timeliness of the present review, the initial screening included studies from the year 2000 onward. However, based on a preliminary bibliometric analysis of publication trends (described in the “Data Extraction and Analysis” section), the final selection was narrowed to studies published between 2019 and 2024, a period marked by a significant increase in PD metabolomics research.Relevance: The primary objective was to include papers that specifically investigate the role of metabolomics in PD. This criterion ensures that the selected studies directly contribute to the understanding of how metabolomics are implicated in the pathogenesis, diagnosis, or treatment of PD. Papers that do not focus on these specific aspects of PD were excluded.Study design: To maintain scientific rigor and reliability, only original research studies and clinical trials involving more than 50 patients with PD were included. This threshold was used to reduce the influence of underpowered analyses and enhance the reliability of reported metabolomic trends in the context of Parkinson’s disease.

Exclusion criteria comprised review articles, letters, conference proceedings, meeting abstracts, case reports, personal communications, and studies involving non-human subjects.

### 3.2. Data Extraction and Analysis

All three query results were merged using the Biblioshiny package in the RStudio (version 4.4.3) environment for Bibliometrix analysis [39]. No records were retrieved from registers. Duplicate entries were first removed using automated tools. Subsequently, an automated filter was applied to exclude studies not written in English and those that did not qualify as original research articles, such as reviews, editorials, or conference proceedings. Prior to manual review, a bibliometric analysis was performed to assess the temporal distribution of the remaining publications in the field of PD and metabolomics. This analysis revealed a marked increase in publication output starting in 2019 (Appendix A). Based on this observation, and to ensure thematic relevance and currency, the review was limited to studies published between 2019 and 2024. This time frame was chosen for two primary reasons. First, the surge in research activity during this period reflects increased attention to metabolomics in PD. Second, focusing on the most recent literature allowed us to include studies employing updated analytical techniques and statistical approaches, thereby improving the coherence and comparability of the included works.

Following this automated filtering, the 80 remaining records were independently screened by two reviewers (F.C. and K.K.K.), who assessed titles, abstracts, and full texts against the predefined eligibility criteria. Any disagreements were resolved through discussion, with the involvement of a third reviewer (N.Z.) when necessary. Articles were excluded if they did not focus on metabolomics, were not correctly indexed as reviews and passed the initial filter, involved fewer than 50 participants, had non-comparable study designs, or were based on non-human models. Studies meeting all inclusion criteria were incorporated into the systematic review [40,41,42,43,44,45,46,47,48,49,50,51,52,53,54,55]. From each included article, the following data were extracted: publication year, journal, authorship, geographic location, study title, design, objectives, inclusion/exclusion criteria, sample size, study timing, metabolomics platforms, statistical approaches, outcomes, main results, and association measures.

### 3.3. Quality Assessment

The Newcastle–Ottawa Scale (NOS) scoring for the selected studies provided an evaluation of their methodological quality based on selection, comparability, and outcome assessment [56,57]. Under this system, each study could earn up to 9 stars. Studies scoring 8 or 9 stars were considered be at low risk of bias, studies that scored 7 or 6 stars were classified at medium risk, and those scoring 5 or less were at high risk.

## 4. Results

### 4.1. Study Selection

A total of 540 records were initially identified through three scientific databases. After removing duplicates, non-English publications, and non-article document types, 120 unique records remained. Of these, 80 full-text articles published between 2019 and 2024 were assessed for eligibility. Following detailed evaluation, 16 studies met all inclusion criteria and were included in the final systematic review (Figure 1).

### 4.2. Characteristics of the Included Studies

The main characteristics of the studies selected for inclusion in this review, consistent with its objectives, are summarized in Table 1. The research studies recruited patients from different countries: USA [41,48,49], China [45,50,54,55], Jordan [47,51], UK [44,53], Luxembourg [52], Germany [40], and India [42]. One study recruited patients both from Germany and USA [43], another study from USA and Italy [46]. This diversity enhances the generalizability of findings but also introduces potential population-specific variability in metabolic profiles.

The age range of participants varied across studies, typically including individuals in their mid-50s to late 60s, consistent with the epidemiology of Parkinson’s disease [1,2,3,4]. Most studies featured a relatively balanced representation of male and female participants, aligning with known prevalence patterns. Sample size varied considerably, from small cohorts (~50 subjects) to large population-based studies involving over 600 PD patients, supporting both exploratory and confirmatory analyses.

Among the selected studies, plasma was the most frequently analyzed biofluid, used in 44% of the cases [40,41,45,52,53,54,55]. This was followed by serum, analyzed in 31% of studies [46,47,48,49,51], and cerebrospinal fluid (CSF) [41,43,48], examined in 19%. Other biofluids, including saliva (6%) [42], urine (6%) [50], and sebum (6%) [44], were less commonly investigated, suggesting their application in more specific research contexts. This variety of sample types enabled exploration of central and peripheral metabolic changes, as well as gut–brain axis involvement.

### 4.3. Research Designs

The studies employed various research designs to ensure robust and meaningful conclusions. Case-control studies [41,42,45,47,48,50,51,55] are the most common, allowing researchers to examine metabolic variations among affected and healthy patients. Observational [44,54] and longitudinal [43] studies are also used, enabling the identification of biomarkers associated with disease progression. Other study designs include cohort-wide profiling [52], metabolome-wide association studies [49], translational studies [46], and integrative analyses [40], indicating a multidimensional approach to metabolomics research. The reviewed literature is predominantly composed of case-control studies, reflecting a primary focus on distinguishing PD patients from healthy controls. Only one longitudinal study was identified [43], highlighting the current limitations in capturing metabolic dynamics over time. Future research should emphasize longitudinal approaches to better understand disease progression and response to treatment.

### 4.4. Risk of Bias

The risk of bias assessment using the Newcastle–Ottawa Scale (NOS) revealed that the majority of included studies demonstrated strong methodological quality (Table 2). Regarding the selection criteria, most studies scored highly in terms of representativeness of the exposed cohort and ascertainment of exposure (9 out of 17 studies scored 4 points in the Selection category). This suggests that patient selection was generally well-documented and adequately representative of the target population. However, the selection of an external control group was often rated as inadequate, suggesting that several studies either failed to define a control group clearly or did not use one appropriately. Without a well-defined and properly implemented external comparison group, there is an increased risk of selection bias, which can compromise the reliability and generalizability of the study findings.

In terms of comparability, only a subset of studies accounted for multiple confounding factors. Studies that scored two points in this category demonstrated strong methodological rigor, improving the reliability of their conclusions. Conversely, those that only scored one point may have lacked adequate adjustments, which could impact the robustness of the associations reported.

The assessment of outcomes was generally well-executed across the studies, with most employing reliable methods for measuring biomarkers and applying statistical validations. However, follow-up duration and the adequacy of follow-up varied significantly among the studies. Those with longer follow-up periods and complete patient tracking provided stronger insights into the stability of biomarkers and disease progression [43,44,47,51]. In contrast, studies with shorter follow-up periods or missing follow-up data were limited in their ability to evaluate long-term biomarker effectiveness [40,41,42,45,46,48,49,50,52,53,54,55].

Overall, eleven studies were classified as low risk of bias (scores of 8–9), demonstrating methodological strength across all three domains. These studies maintained rigorous selection criteria, adjusted for multiple confounders, and implemented sufficient follow-up durations to enhance the reliability of their findings. The remaining five studies, scoring between six and seven out of nine, were generally sound in their methodology but exhibited weaknesses in either confounder adjustments or follow-up duration. Importantly, none of the reviewed studies scored below six, indicating that all studies adhered to a reasonable methodological standard.

### 4.5. Confounding Variables: Sex, Age, Medications, Diet

Controlling for confounding variables is a critical aspect of study design and analysis in metabolomics, particularly in complex and heterogeneous conditions such as Parkinson’s disease. Factors including age, sex, medication use (especially levodopa), comorbidities, and dietary habits can significantly influence the metabolomics profile of patients, potentially obscuring disease-specific alterations [58]. Across the studies reviewed, the degree to which these confounders were documented, measured, and statistically adjusted for varied widely.

#### 4.5.1. Sex and Age

Epidemiological evidence confirms significant sex differences in PD incidence and clinical presentation with men showing nearly twice the risk compared to women [58]. Risk also increases with age, though the precise nature of this trend remains unclear [58]. These observations underscore the importance of investigating the biological mechanisms underlying sex- and age-specific differences in PD pathogenesis, especially within the framework of precision medicine. Most of the reviewed studies ensured demographic comparability between PD patients and controls through matching for sex and age during recruitment [40,41,46,47,48,50,51,55], with some adding further statistical adjustments [49,52,53]. Only a minority explicitly investigated sex as biological modifiers of the metabolome. Significant sex differences were identified in circulating fatty acids and their conjugates [54]. For instance, female PD patients exhibited higher levels of unsaturated fatty acids, potentially reflecting lifelong lower lipase expression compared to males. On the contrary, most glycerophosphocholines were decreased in females, suggesting sex-specific disruptions in membrane lipid dynamics and metabolism. Another study found that metabolic differences between male and female controls were attenuated in PD patients, hypothesizing a potential “metabolic reprogramming” driven by disease progression [45]. No evidence of sex-related confounding in salivary [42] and sebum [44] metabolomic data were observed. None of the reviewed studies assessed the impact of age on the PD metabolome.

#### 4.5.2. Dopaminergic Therapies

The widespread use of dopaminergic drugs, particularly levodopa, requires careful consideration of their impact on the PD metabolome, as treatment status and drug exposure can markedly influence metabolic profiles and confound disease-specific changes. To minimize this effect, some studies focused on drug-naïve de novo patients [52] or collected samples during medication withdrawal (“off-state”) [40]. In early-stage PD, systematic assessment of metabolite– Levodopa Equivalent Daily Dose (LEDD, a standardized measure that converts different antiparkinsonian medications into an equivalent daily dose of levodopa) relationships showed that drug-derived metabolites, mainly linked to dopamine metabolism, clustered together but correlated weakly with overall PD-related changes [49]. A multicenter study further reduced pharmacological interference by documenting LEDD and excluding drug-related metabolites from biomarker pipelines [48].

Incorporating LEDD as a covariate in statistical models showed that dopaminergic therapy significantly increased levels of 3-methoxytyrosine in CSF [43] and plasma [45]. Notably, no metabolites were associated with pramipexole monotherapy, but three compounds, namely phenylalanine, L-3-methoxytyrosine, and phosphatidylethanolamine O-38:6, were linked to combination therapy, suggesting synergistic effects on specific metabolic pathways [45]. In salivary metabolomics data, LEDD moderately correlated with histidine, tyrosine, and phenylalanine, all of which were elevated in patients compared to controls [42].

#### 4.5.3. Diet

Diet plays a key role in Parkinson’s disease by influencing risk, progression, and treatment response [59]. Adherence to the Mediterranean or Mediterranean–DASH Diet Intervention for Neurodegenerative Delay (MIND) diets, both rich in plant-based foods and healthy fats, has been associated with a reduced risk of PD, whereas high intake of saturated fats and sugars may promote neurodegeneration. Nutrients like polyphenols, omega-3 fatty acids, and fiber help regulate inflammation, oxidative stress, and gut microbiota. Protein can affect levodopa absorption, and B vitamins may lower homocysteine levels.

Given the central role of metabolism in PD pathophysiology, dietary interventions are also likely to influence the metabolomic profile, through both direct effects of nutrient intake and indirect effects mediated by the gut microbiota. Among the studies included in this review, only one explicitly investigated the impact of diet on metabolomics [52]. The findings reported alterations in metabolites typically associated with plant-based diets (oxalate, tartronate) and microbial activity (4-hydroxyphenylacetate, butyrate) in PD patients, particularly in the de novo subgroup, suggesting a possible contribution of dietary habits and gut microbiota to metabolic differences between PD patients and controls.

### 4.6. Analytical Platforms and Techniques

A variety of analytical platforms are utilized to investigate metabolic profiles. Liquid Chromatography-Mass Spectrometry (LC-MS) emerges as the most widely used technique [44,47,49,50,51,52,54,55], known for its sensitivity and broad metabolite coverage. Other advanced techniques include Nuclear Magnetic Resonance (NMR) spectroscopy [42,46,53], Ultra-High-Performance Liquid Chromatography-Mass Spectrometry (UHPLC-MS/MS) [48], Gas Chromatography-Time of Flight Mass Spectrometry (GC-TOF-MS) [51], and GC-MS [40]. The diversity of techniques reflects the need for complementary approaches to capture a comprehensive metabolic profile. The selection of the analytical method depends on the study objectives, available resources, and the specific metabolites of interest. Combining MS and NMR techniques can maximize metabolite coverage, enhance metabolite identification, and improve data reliability, offering a more holistic view of the metabolome [60,61].

### 4.7. Statistical Methods

The analysis of high-dimensional metabolomics data typically involves a range of statistical approaches aimed at identifying discriminant metabolites, understanding biological variation, and developing predictive models. However, the complexity and high dimensionality of metabolomics datasets present unique challenges.

Univariate statistical tests remain useful for identifying individual metabolites that differ significantly between groups, but they may overlook complex multivariate interactions. Multivariate techniques, such as Principal Component Analysis (PCA), Partial Least Squares Discriminant Analysis (PLS-DA), and Orthogonal PLS-DA (OPLS-DA), are commonly used for data reduction and pattern recognition. PCA, an unsupervised technique, is valuable for exploratory data analysis and dimensionality reduction, but it does not account for group classification and may fail to highlight disease-specific features. PLS-DA, a supervised method, enhances class discrimination but is prone to overfitting, especially when the number of variables greatly exceeds the number of samples, a common scenario in metabolomics. PLS-DA models should always be validated through permutation tests and cross-validation to ensure robustness. To assess the diagnostic performance of metabolite-based models, Receiver Operating Characteristic (ROC) curve analysis is commonly used. One of its key metrics, the Area Under the Curve (AUC), provides a straightforward measure of a model’s ability to discriminate between disease and control groups. AUC values range from 0.5 (no discrimination) to 1.0 (perfect classification), making them an intuitive and powerful indicator of predictive accuracy.

In the reviewed studies, univariate group comparisons were often conducted using parametric tests such as Welch’s *t*-test [40], Student’s *t*-test [47,48,55], and ANOVA [46,47,51], alongside nonparametric Mann–Whitney U tests [41,42,45,55] to identify significantly altered metabolites. Exploratory unsupervised methods, principal component analysis (PCA) [45,54] and hierarchical clustering [45], show how biospecimens naturally group based on their metabolic signatures. PLS [48], PLS-DA [42,44,47,49,50,51,55], and OPLS-DA [42,45,46,54] were leveraged for dimensionality reduction and classification. Predictive modeling strategies span classical regressions (linear regression [43,49,52]; logistic [49,53] and binary logistic regression [45]) and machine learning algorithms, including Support Vector Machines [40,48,52,55], Random Forests [40,50,55], XGBoost [50], k-nearest neighbors [55], and penalized methods (LASSO [48,50,55], ridge regression [50]). Longitudinal cohort analyses utilized linear mixed-effects models [43] to account for repeated measures, and survival outcomes have been assessed via Cox proportional hazards modeling [53]. Across studies, ROC curve analysis [40,41,42,43,44,45,46,50,52,53,55] has served as the principal metric of diagnostic and prognostic performance, with complementary pathway enrichment [44,48,52] and ChemRICH network analyses [51], providing biological context to the observed metabolic alterations.

### 4.8. Biofluid-Specific Metabolomics Signatures

The selection of biofluids in metabolomics plays a pivotal role in capturing disease-related metabolic alterations, as each matrix offers unique biochemical perspectives. In Parkinson’s disease, a variety of biofluids have been investigated, each of them differing in terms of accessibility, proximity to the central nervous system, and potential for biomarker discovery. This section presents a synthesis of key findings from the selected studies, organized by biofluid type, and highlights the associated metabolic changes. The aim is to identify candidate biomarkers that may support early diagnosis, monitor disease progression, and inform therapeutic strategies.

#### 4.8.1. Plasma: Systemic Indicators of Oxidative Stress and Lipid Dysregulation

Plasma is the most widely used biofluid in PD metabolomics studies, comprising approximately 44% of analyzed research [40,41,45,52,53,54,55]. Its widespread use is due to its accessibility and ability to reflect systemic metabolic changes.

Plasma-based metabolomics studies highlighted significant disturbances in lipid metabolism, amino acid metabolism, and oxidative stress markers in PD patients. Different lipid classes among which fatty acids, glycerophosphocholine, triacylglycerides, and lysophostatidylcholines showed significant differences between PD patients and controls [40,45,53,54,55], with metabolite-specific trends and marked sex-related variability [54]. Several metabolites related to fatty acid metabolism, particularly mitochondrial β-oxidation, were also altered in PD plasma [52]. Benzoylcarnitine, an acylcarnitine involved in transporting fatty acids into mitochondria, showed reduced levels in PD patients compared to controls, consistent with impaired β-oxidation [52]. In de novo PD, the 2-butenoylglycine content was elevated but not when comparing treated PD to controls. The latter, an acylglycine produced via glycine conjugation of crotonyl-CoA, may reflect a detoxification response to its accumulation and has been linked to both mitochondrial β-oxidation and bacterial butyrate production, suggesting a shared pathway alteration [52]. A significant increase in butyrate, a further metabolite associated with fatty acid β-oxidation and formed by gut bacteria, was also noted in de novo PD vs. control whereas this difference was not observed in treated PD [52].

Altered levels of amino acids including tyrosine, tryptophan, and valine were observed in plasma [41,53]. Markers of oxidative stress, including disrupted xanthine metabolism, further supported the involvement of mitochondrial dysfunction, a well-established hallmark of PD [40,52]. In addition, dysregulation of the kynurenine pathway, the main catabolic route of tryptophan, was indicated by altered concentrations of multiple kynurenine-derived metabolites [41,45].

Notably, circulating xanthine, 2-ketocaprylate (an intermediate of branched-chain amino acids), and glutarylcarnitine emerged as top discriminant metabolites for distinguishing de novo PD patients from healthy controls [52]. These markers performed well in machine learning models, with xanthine showing the best results, reaching AUC values around 0.71–0.72, indicating good diagnostic potential.

Another biomarker panel of fourteen metabolites identified in plasma was specifically linked to existing PD, including amino acids (like tyrosine and valine), fatty acids such as omega-3, docosahexaenoic acid (DHA), and total fats, and various lipoprotein types [53]. Researchers compared three models to predict the risk of developing Parkinson’s within 10 years: one based on traditional risk factors, one based on metabolites, and a combined model. The traditional model showed good predictive performance (AUC = 0.766), while the metabolite-only model was less accurate (AUC = 0.580). Adding metabolites to the traditional model slightly improved prediction (AUC = 0.768), but the improvement was not statistically significant.

Notably, combining positron emission tomography (PET) imaging with plasma metabolomics showed strong potential for improving clinical decision-making in Parkinson’s disease [40]. The highest predictive accuracy was achieved when standardized PET imaging features were integrated with metabolomics data. In particular, a support vector machine (SVM) model reached an AUC of 0.98 when using fluorodopa (FDOPA) PET imaging combined with metabolomics, and an AUC of 0.91 for fluorodeoxyglucose (FDG) PET combined with metabolomics. By comparison, models based only on PET imaging performed less well (FDOPA: AUC = 0.94; FDG: AUC = 0.80), and metabolomics alone had substantially lower accuracy (AUC = 0.66). These findings highlight the added value of integrating imaging and molecular data for more accurate disease classification.

#### 4.8.2. Serum: Lipid Dysregulation and the Gut–Brain Axis

Serum metabolomics accounts for 21% of PD studies included in this review. While partially overlapping with plasma-based findings, serum analyses provided additional insights, particularly into lipid metabolism and gut–brain axis interactions. Altered levels of lysophosphatidylcholines were observed in PD subjects, suggesting compromised membrane integrity and ongoing neurodegenerative processes [47,49,51]. Three metabolite panels showed strong diagnostic potential: a lipid-based panel of ten compounds (AUC = 0.974) [47]; a seven-metabolite set, including cysteine-S-sulfate, 1-methylxanthine, vanillic acid, *N*-acetylaspartic acid, 3-*N*-acetyl tryptophan, 5-methoxytryptophol, and 13-hydroxyoctadecadienoic acid, namely 13-HODE (AUC = 0.977) [51]; a composite profile of six metabolites including acetoacetate, betaine, b-hydroxybutyrate (BHB), creatine, pyruvate, and valine (AUC = 0.88) [46].

Several gut microbiota-derived metabolites were consistently elevated in PD, including phenylacetylglutamine, produced from phenylalanine and tyrosine putrefaction, and p-cresol along with its conjugates p-cresol sulfate and p-cresol glucuronide [49]. In parallel, glutamate and related metabolites (e.g., pyroglutamic acid) were also more abundant in PD, with glutamine/glutamate metabolism significantly overrepresented.

Additional alterations included branched-chain amino acids contents, pointing to impaired energy metabolism [46,49], and purine metabolism abnormalities, supporting the role of oxidative stress and mitochondrial dysfunction in PD progression [51]. Ten compounds, including *N*-acetylisoleucine, phenylacetylcarnitine, and phenylacetylglutamine, showed great associations with PD diagnosis; however, none retained statistical significance after adjustment [48].

#### 4.8.3. Cerebrospinal Fluid (CSF): Direct Insights into Neurochemical Alterations

CSF is the biofluid that most directly reflects the biochemical state of the central nervous system, making it particularly valuable for studying neurodegenerative disorders like PD. In early-stage PD patients, significantly reduced levels of two dopamine metabolites, namely homovanillic acid (HVA) and 3,4-dihydroxyphenylacetic acid (DOPAC), were observed compared to healthy controls, confirming the central role of dopaminergic degeneration in the initial phases of the disease [43]. Remarkably, these baseline levels remained stable over a two-year period in drug-naïve patients, with changes occurring only after levodopa treatment initiation.

Polyamine metabolism also appears to contribute to PD neurodegeneration. By integrating data from LASSO regression and univariate analysis, eleven compounds were identified as significantly distinguishing PD patients from controls. Among these, *N*-acetylcadaverine and *N*-acetylputrescine exhibited the most pronounced fold-changes [48]. For the subset of metabolites selected, the area under the ROC curve reached 0.897, indicating strong discriminatory power.

Disruption of tryptophan metabolism, particularly the kynurenine pathway, was also evident. CSF levels of kynurenic acid, a neuroprotective metabolite, were 23% lower in PD patients compared to controls [41]. In contrast, higher levels of quinolinic acid, a compound that can damage the nerve cell, were associated with more severe clinical symptoms. Notably, reduced kynurenic acid levels were also linked to olfactory deficits, an early non-motor symptoms of PD.

#### 4.8.4. Saliva: A Non-Invasive Avenue for Biomarker Discovery

Saliva-based metabolomics represents a promising, non-invasive diagnostic tool for PD, revealing amino acid imbalances, ketone body variations, and microbial metabolite shifts [42]. In PD patients, fifteen metabolites were found to be more abundant compared to healthy controls, including phenylalanine, tyrosine, histidine, glycine, γ-aminobutyric acid (GABA), *N*-acetylglutamate (NAG), acetoacetate, acetoin, acetate, alanine, fucose, propionate, isoleucine, valine, and trimethylamine-*N*-oxide (TMAO). Increase in neurotransmitter-related amino acids (GABA, phenylalanine, tyrosine, and histidine) suggested altered neurotransmitter processing, while elevated TMAO levels pointed to gut microbiota dysfunction. Changes in ketone bodies (acetoin and acetoacetate) were consistent with adaptive energy metabolism. Notably, a subset of these metabolites, including histidine, propionate, tyrosine, isoleucine, acetoin, NAG, acetoacetate, and valine, showed good diagnostic accuracy, with AUC values ranging from 0.67 to 0.72 (95% CI). Despite its diagnostic potential, salivary metabolomics remains an underexplored field and requires further validation for clinical application.

#### 4.8.5. Sebum: A Novel Lipidomic Fingerprint for PD

Sebum metabolomics offers a unique approach to PD biomarker research, focusing on lipidomic profiling. Its non-invasive nature and the distinct lipidomic alterations observed in PD patients make it a promising candidate for diagnostic applications, revealing ceramide and fatty acid imbalances. Altered ceramide levels, particularly hexosylceramides (HexCer), Cer(42:0), and Cer(40:0), indicated disruptions in lipid signaling pathways linked to neurodegeneration [44]. Observed reduction in triglycerides, including TG(50:5), further suggested an alteration in overall lipid balance. However, the standardization of sampling techniques remains a challenge in establishing sebum-based diagnostics.

#### 4.8.6. Urine: A Metabolic Readout of Systemic Contributions to Neurodegeneration

Urine metabolomics offers insights into systemic metabolic dysfunction and host–microbiome interactions. The Ingenuity Pathway Analysis (IPA) of urinary metabolites altered in PD highlighted the involvement of three metabolites in key signaling pathways and molecular networks [50]. In particular, altered metabolites such as 3-methoxytyramine, xanthine, uric acid, and vanillic acid were connected to major canonical pathways, including dopamine degradation, dopamine receptor signaling, neuroinflammation signaling, and mitochondrial dysfunction. Uric acid, an antioxidant from purine metabolism, was linked to slower disease progression; 3-methoxytyramine reflects reduced dopamine release [62]; and vanillic acid, a catecholamine byproduct, has shown antioxidant properties and neuroprotective effects in rotenone-induced PD models, mitigating behavioral, histopathological, and neuroinflammatory changes [63].

### 4.9. Integration with Other Omics Approaches

While metabolomics alone offers a detailed snapshot of the biochemical alterations associated with PD, its full potential emerges when integrated with other omics platforms, such as genomics, transcriptomics, proteomics, lipidomics, and microbiomics, within a multi-layered analytical framework. Each omics discipline captures a distinct layer of biological organization: genomics identifies inherited and somatic variants that may predispose individuals to PD; transcriptomics reflects dynamic gene expression changes; proteomics reveals protein abundance and post-translational modifications; lipidomics offers insights into membrane composition and lipid signaling; microbiomics characterizes gut microbial communities and their metabolic output. Combining these complementary datasets with metabolomics enables the identification of cross-level interactions, such as linking metabolic profiles to genetic polymorphisms, mapping protein–metabolite networks, and associating gut microbiota shifts with circulating or CNS-specific metabolites.

Among the studies selected for this review, only one applied an integrative framework [52]. This analysis highlighted xanthine metabolites ( inosine, xanthosine, xanthine, and hypoxanthine) as highly significant and predictive for PD, showing a consistent pattern of increased abundance in de novo patients compared to controls. The metabolic profile was mechanistically aligned with transcriptomics data, revealing reduced expression of hypoxanthine–guanine phosphoribosyl transferase (HPRT1), the enzyme that converts hypoxanthine to inosine monophosphate. This enzymatic downregulation offers a plausible explanation for xanthine accumulation and suggests a downstream shortage of cellular ATP as a pathological consequence.

Although excluded from our selection due to participant number criteria, other studies have also demonstrated the value of integrated omics approaches in PD research. For example, in patients with TMEM175 mutations, combined lipidomics, metabolomics, and proteomics of plasma and fibroblasts revealed lysosomal, autophagy, and mitochondrial dysfunction, highlighting the genetic basis of multi-level molecular changes in PD [64]. Metabolome–microbiome integration uncovered PD-specific microbial–host sulfur co-metabolism patterns potentially affecting disease severity [65]. Proteomics–metabolomics integration confirmed lipid metabolism disturbances as a central pathogenic feature, with activation of the sphingolipid pathway and reduced apolipoproteins [66].

Collectively, these findings illustrate the power of multi-omics strategies not only to capture biologically relevant changes, but also to validate findings across different biological domains. Expanding the use of such integrative approaches in larger, well-characterized cohorts could accelerate the discovery of mechanistically informed biomarkers and therapeutic targets.

## 5. Discussion

This review provides an updated synthesis of recent metabolomics studies on Parkinson’s disease, focusing on publications from the past five years that examined large human cohorts. Building on previous literature, it underscores the complexity and heterogeneity of metabolic alterations in PD by comparing findings across multiple biological matrices and metabolic pathways. While plasma and serum remain the most frequently analyzed biofluids due to their ease of collection and systemic significance, the inclusion of cerebrospinal fluid (CSF), saliva, sebum, and urine in recent studies has expanded the analytical framework, offering complementary perspectives on central, peripheral, and microbiota-related metabolic disruptions. Table 3 summarizes the major metabolic alterations identified in PD across different studies and biofluids.

### 5.1. Amino Acid Metabolism

Amino acid metabolism is consistently altered across serum [46,49], saliva [42], and plasma [41,53] in PD. Changes involve both aromatic amino acids (e.g., tyrosine, phenylalanine) and branched-chain amino acids (BCAAs), namely valine, isoleucine, and leucine, along with neurotransmitter precursors. In plasma, tyrosine, a precursor for dopamine, and tryptophan, a serotonin precursor metabolized through the kynurenine pathway [67,68], showed notable alteration. In serum samples, elevated levels of glutamate, phenylacetyl-L-glutamine, pyroglutamic acid, serine, and isoleucine suggested altered protein turnover and dysfunction in neurotransmitter biosynthetic pathways. Glutamate increases, in particular, was associated with excitotoxicity, which exacerbates neuronal damage [69,70]. BCAAs are crucial for mitochondrial function and energy metabolism [71]. In saliva, higher N-acetylglutamate, glycine, and GABA point to neurotransmitter dysregulation.

### 5.2. Lipid Metabolism

Lipids play critical roles in cell membrane integrity, supporting myelin sheath maintenance, and regulating intracellular signaling pathways. Alterations in lipid profile are strongly involved in PD, consistent with the well-established role of lipids in the pathogenesis of neurodegenerative diseases. Although trends may differ among metabolites within the same molecular class, changes in sphingolipids, phosphatidylserines, ceramides, and free fatty acids were observed across various biofluids such as plasma [40,45,53,54,55], serum [47,49,51], and sebum [44].

Alteration in ceramides, sphingomyelins, and phosphatidylserines which are integral components of cellular membranes and lipid-mediated signaling, in serum [47] and sebum [44], potentially reflect membrane instability and neuroinflammatory processes. Changes in glycosphingolipids [44], lysophosphatidylcholines [47,49], and fatty acyl metabolites [44] pointed to compensatory remodeling of membrane lipids. Elevated levels hexadecenoic acid and dodecanoic acid in plasma [40], and several other free fatty acids in serum [49], along with increased unsaturated triacylglycerides [47], suggested disruptions in fatty acid metabolism, particularly β-oxidation [72]. Decreased glycerolipids and phospholipids in plasma [55] and reduced triacylglycerols in sebum [44] were linked to impaired lipid storage and transport.

Notably, significant sex-specific differences in lipid metabolism were also observed in PD patients [54]. For instance, male patients showed marked alterations in eicosanoid profiles, while female patients exhibited enrichment of glycerophosphocholines, most of which appear downregulated compared to healthy controls. These findings point to a complex and heterogeneous pattern of differentially expressed lipid metabolites, which may reflect underlying sex-dependent metabolic reprogramming in Parkinson’s disease.

### 5.3. Energy Metabolism

Closely linked to these lipid disturbances is mitochondrial dysfunction, a central pathological mechanism in PD [73,74]. Altered levels of citric acid [55], pantothenic acid [49], alanine [42], orotic acid [50], 3,3-dimethylglutaric acid [50], and oxoglutaric acid [49] suggest impaired mitochondrial energy production. The increase in ketone bodies observed in saliva [42] and serum [46] point to a compensatory ketogenesis due to impaired glucose metabolism [40]. These metabolic changes, observed across multiple biofluids, indicate a shift toward alternative energy sources in response to mitochondrial dysfunction. Together, these findings underscore bioenergetic deficits as a core feature of PD pathology.

### 5.4. Oxidative Stress

Mitochondrial impairment directly contributes to increased oxidative stress, another hallmark of PD. The accumulation of reactive oxygen species (ROS) due to compromised mitochondrial respiration leads to widespread oxidative damage to proteins, lipids, and DNA. Evidence for oxidative stress involvement in PD is underscored by alterations in xanthine and its derivatives in plasma [40,52], serum [51], and urine [50], as well as uric and vanillic acids in plasma [52]. Xanthines, intermediates of purine metabolism, can contribute to oxidative burden when dysregulated. In contrast, uric acid, the final product of purine catabolism, has antioxidant properties and may help counteract oxidative stress. Similarly, vanillic acid, a catecholamine metabolite, exhibits antioxidant and neuroprotective effects.

### 5.5. Neurotransmitter Metabolism

Alterations in the tryptophan–kynurenine pathway point to a dysregulation of tryptophan catabolism in PD [67,68]. They included reduced kynurenine levels in plasma [45] and serum [49], decreased 3-hydroxyanthranilic and kyrunenic acid in plasma [41], lower quinolinic acid in CSF [41], increased 3- hydroxykynurenine in plasma [54], and decreased indolelactic acid in plasma [45]. This pathway is critical for both mitochondrial energy metabolism and the synthesis of neuroactive compounds, including serotonin and melatonin, and its imbalance can lead to increased production of neurotoxic metabolites (e.g., quinolinic acid) or reduced levels of neuroprotective ones (e.g., kynurenic acid). Such disturbances are associated with excitotoxicity, oxidative stress, and neuroinflammation, hallmarks of PD pathology.

Evidence of compensatory modulation of neurotransmitter systems in PD subjects emerged from salivary analyses, which revealed significant increases in neurotransmitter-related metabolites such as N-acetylglutamate (NAG), glycine, and γ-aminobutyric acid (GABA) [42].

Alterations in dopamine metabolism are also evident in PD, as shown by increased urinary and CSF levels of 3-methoxytyramine and *N*-acetyl-1-tyrosine, alongside decreased concentrations of 3,4-dihydroxyphenylacetic acid (DOPAC) and homovanillic acid (HVA) which are major end-products of dopamine degradation [43,50]. The reduction in DOPAC and HVA, both major end-products of dopamine degradation, suggests impaired dopamine turnover, consistent with the loss of dopaminergic neurons in the substantia nigra. Conversely, the increase in 3-methoxytyramine, a product of dopamine O-methylation, was linked to compensatory changes in catechol-O-methyltransferase (COMT)-mediated metabolism.

### 5.6. Gut Microbial Metabolites

Gut ecosystem imbalance also plays a key role in the pathogenesis of PD, contributing to gut–brain axis disruption, blood–brain barrier instability, and ultimately the neuronal degeneration [75,76]. Serum, saliva, and plasma metabolomics analyses revealed the production of abnormal amounts of microbial metabolites such as p-cresol, p-cresol sulfate, and p-cresol glucuronide [45,49,53], TMAO [42], and phenylacetyl-glutamine [49]. These findings reinforce the emerging view that PD is not merely a disorder of the central nervous system but a disease with significant peripheral metabolic components, in which gut-derived metabolites may act as modulators of neurodegeneration.

### 5.7. Polyamine Metabolism

Finally, disruptions in polyamine metabolism, involving compounds such as *N*-acetylputrescine and *N*-acetylcadaverine, detected in CSF [48] and serum [51], add a further layer of complexity. Polyamines are involved in cell growth, immune signaling, and ion channel function, and their dysregulation has been linked to oxidative stress, mitochondrial dysfunction, and α-synuclein aggregation, further tying this pathway to the multifactorial pathology of Parkinson’s disease [77].

### 5.8. Metabolome Association with PD Stage and Duration

PD duration and clinical staging, typically assessed by the Hoehn and Yahr (H&Y) scale [78], influence both the severity and nature of neurodegeneration, and are therefore likely to influence the associated metabolomics profile. Understanding these alterations is essential for identifying reliable biomarkers for diagnosis, progression, and treatment response. Stratifying patients by these parameters can improve biological interpretability, helping to differentiate early pathogenic processes from downstream metabolic effects of disease progression or therapy. Notably, only a subset of the reviewed studies explicitly considered disease stage or duration, yet these provided valuable insights into how metabolomics signatures change along the clinical trajectory of PD.

In one study, PD patients were stratified into early (H&Y ≤ 2) and advanced (H&Y > 2) stages to examine the salivary metabolome [42]. Butyrate levels showed a negative correlation with H&Y stage (r = −0.25, *p* = 0.04), suggesting a decline with increasing disease severity. In contrast, propionate and acetoin concentrations were positively correlated with disease duration, but not with H&Y stage or motor scores, indicating that some metabolic changes may be more closely linked to illness duration than to clinical staging. These findings point to long-term alterations in host–microbiome interactions as a feature of PD progression.

A comprehensive study examined serum and tissue samples from three animal models and two human cohorts to identify metabolic changes associated with different stages of Parkinson’s disease [46]. In the early, prodromal-like phase, levels of alanine, BHB, glycine, lactate, and serine were elevated, while betaine was reduced. These alterations remained stable as the disease progressed. Notably, pyruvate levels showed a steady increase throughout disease advancement. In the clinical-like stage, dimethyl sulfone (DMSO_2_) and valine decreased, whereas threonine increased. Across all animal and human models, six metabolites (acetoacetate, betaine, creatine, BHB, pyruvate, and valine) were consistently dysregulated. A logistic regression model built on these biomarkers achieved remarkable predictive performance: 92.5% accuracy for PD stage classification in animals (AUC = 0.936), 82.6% accuracy in a human cohort (AUC = 0.83), and an 84.6% classification rate in a second human cohort.

In plasma-based metabolomics, patients assessed at H&Y stages 2–4 showed distinct associations between metabolite levels and disease severity [55]. Specifically, phosphatidylcholines, such as PC (40:7), and eicosatrienoic acid negatively correlated with PD severity, whereas pentalenic acid, PC (40:6p), and aspartic acid were positively associated. These metabolites were incorporated into a multi-analyte predictive model, achieving a diagnostic accuracy greater than 81.6%.

Another study explored the associations between plasma metabolites and clinical measures including H&Y stage, Unified Parkinson’s Disease Rating Scale (UPDRS), and LEDD [49]. Among the metabolites analyzed, p-cresol glucuronide was positively associated with higher H&Y stage (FDR = 0.095), while 3-O-methyldopa, a metabolite related to PD medication, showed significant correlations with both H&Y and UPDRS-III. Although additional metabolic features evidenced statistical associations with disease stage or severity, most lacked confident annotation, highlighting the need for improved metabolite identification in future studies.

Serum lipidomics analyses comparing early- and advanced-stage PD patients revealed differences in several molecular classes [47,51]. The level of cysteine-S-Sulfate, N8-acetyl spermidine, 13-HODE, and galactosamine-1-phosphate increased with disease progression, while the content of lysophosphatidylinositol (LPI) 20:4 was significantly decreased [51]. Lysophosphatidylcholine (LPC) O-18:1, 12-hydroxyeicosatetraenoic acid (12-HETE), and phosphatidylethanolamine (PE) 36:0 levels were significantly elevated in PD patients versus controls and continued to rise with disease severity [45].

Levels of plasma 3-HK were significantly correlated with PD duration [41]. 3-HK also correlated with clinical severity (UPDRS I–III). These associations support a role for oxidative stress and inflammation in PD progression and point to 3-HK as a potential biomarker of disease duration and severity.

## 6. Strengths, Limitations, and Translational Outlook of Current Metabolomics Research in Parkinson’s Disease

This review has several strengths, including literature search, clearly defined inclusion criteria, and the use of a standardized tool (Newcastle–Ottawa Scale) for quality assessment. Most of the included studies were rated as moderate to high quality, particularly in the domains of patient selection and outcome measurement. However, some limitations should be acknowledged in the current body of Parkinson’s disease metabolomics literature.

### 6.1. Methodological Limitations and Confounding Factors

A major challenge is the substantial methodological heterogeneity observed across studies, encompassing demographic composition (e.g., sex and age distributions), dietary influences, pharmacological treatment (especially levodopa use and dosage), and technical aspects of data acquisition and processing. Only a minority of studies adequately addressed confounding factors. The lack of standardized protocols for patient stratification and medication treatment complicates the interpretation of metabolite differences, making it difficult to disentangle disease-specific alterations from those related to confounding variables.

Analytical variability, such as differences in the biofluids, sample collection protocols, metabolite extraction methods, analytical platforms, and statistical workflows, also play an important role in the reproducibility of results across independent cohorts. The inclusion of biofluids, such as saliva, urine, and sebum, suggests promising avenues for the development of non-invasive diagnostic and prognostic tools. However, each matrix has specific drawbacks that may hinder reproducibility and clinical translation. Saliva, for example, is highly sensitive to dietary intake, oral hygiene, circadian variation, and collection protocols, introducing both inter- and intra-individual variability [79]. Sebum analysis suffers from the absence of standardized collection procedures and is influenced by anatomical site, hormonal regulation, and potential contamination from skin microbiota or cosmetic products [80]. Urine, while easily obtainable, is more reflective of systemic and renal physiology than central nervous system activity, and can be affected by hydration status and kidney function [81].

### 6.2. Need for Standardization and Harmonization

All the above-mentioned factors underscore the need for standardized procedures in clinical metabolomics. In particular, harmonization of sample collection, processing, quality control, pre-analytical handling, and data normalization is essential to improve cross-study comparability and enable robust meta-analytical approaches. Addressing these methodological challenges will demand coordinated efforts to define, validate, and implement shared standard operating procedures (SOPs), which are currently lacking [82]. The establishment of such guidelines will be a pivotal step towards enhancing reproducibility, data integration, and ultimately, clinical translation of metabolomics findings.

### 6.3. Translational Outlook and Future Directions

#### 6.3.1. Strengths and Translational Potential of Current Metabolomics Studies

Despite the above-mentioned limits, the studies included in this review display several noteworthy strengths. Some applied patient stratification by disease stage or duration, offering a more refined understanding of how metabolic alterations evolve throughout the course of PD. Many employed advanced statistical and machine learning approaches to develop predictive biomarker panels with high diagnostic accuracy (in some cases achieving AUC values above 0.90), underscoring their potential clinical utility. The inclusion of specific metabolite–clinical measure correlations strengthened the clinical relevance of the results, linking biochemical changes to functional outcomes. Collectively, the identified metabolic alterations identified across studies provide basic knowledge for future clinical translation. While further validation is essential, several metabolites and metabolic pathways have shown consistent and biologically plausible associations with PD pathophysiology. These include disruptions in mitochondrial function, redox homeostasis, lipid metabolism, and microbial-derived metabolites. These recurring findings, thus, reinforce the biological plausibility of metabolomics as a tool for uncovering mechanisms underlying PD onset and progression.

#### 6.3.2. Metabolomics Perspective on Atypical Parkinsonian Syndromes

Distinguishing PD from atypical parkinsonian syndromes (APS), primarily Progressive Supranuclear Palsy–Parkinsonism (PSP-P) and Multiple System Atrophy (MSA), remains particularly challenging in the early disease stages, when clinical features are not yet fully distinctive. PSP-P often presents with asymmetric onset, tremor, and a moderate initial response to levodopa, thereby closely resembling PD during the early disease course [18]. These shared motor features can obscure diagnosis, especially when non-motor signs and disease-specific biomarkers are not yet fully developed. Conversely, distinctive elements such as earlier postural instability, oculomotor dysfunction, or faster disease progression may emerge over time, providing diagnostic clues. Similarly, MSA is difficult to identify early, as it frequently shares symptoms with PD [83,84]. MSA commonly exhibits symmetrical parkinsonian features, and atypical signs, including poor levodopa responsiveness, autonomic dysfunction, cerebellar ataxia, pyramidal signs, and severe immobility, often develop gradually, further delaying accurate diagnosis. Since PD and APS disorders differ markedly in prognosis and therapeutic strategies, accurate differentiation is critical for effective disease management. However, diagnostic accuracy remains suboptimal, with reported rates of only 62–79% for MSA [83] and approximately 80% for both PD and PSP [85]. As current diagnoses still rely predominantly on clinical evaluation, distinguishing among PD, PSP, and MSA continues to pose significant difficulties.

Although none of the studies in this review directly investigated shared and disorder-specific biochemical signatures across APS, metabolomics studies outside our inclusion criteria provided preliminary evidence of the potential of this approach to differentiate these conditions. For example, plasma-based analyses revealed that patients with MAS (*n* = 16) show elevated levels of free fatty acids, suggesting impaired β-oxidation or defective mitochondrial fatty acid transport, while both MSA and PSP (*n* = 20) patients exhibit reduced LysoPC(16:0), indicative of alterations in phosphatidylcholine metabolism [86]. PSP was further characterized by changes in oxidative stress-related markers, including decreased uric acid and elevated cysteine–glutathione disulfide, highlighting a high oxidative stress burden. Comparisons of these findings with those of PD studies on plasma and serum revealed partially overlapping features, such as increased fatty acids and reduced uric acid, suggesting shared pathogenic mechanisms [86].

A targeted study measuring amino acids, acylcarnitines, and lipids in serum and CSF from PD (*n* = 11) and APS (*n* = 8) patients identified consistent alterations in tyrosine, putrescine, trans-4-hydroxyproline, and total dimethylarginine, pointing to disturbances in catecholamine metabolism, polyamine and proline turnover, and nitric oxide pathways [87]. While these features robustly separated patient groups from controls, discrimination between PD and APS was more modest, underscoring the need for larger, harmonized cohorts and multi-matrix designs.

Another metabolomics investigation on PD (*n* = 34), MSA (*n* = 12), and PSP (*n* = 17) patients showed that each condition displays a unique set of discriminatory metabolites capable of separating patients from controls (*n* = 31), highlighting disease-specific metabolic signatures despite shared neurodegenerative mechanisms [88]. Notably, only two metabolites, formic acid and succinate, showed consistent alterations across all three disorders. Both are closely linked to mitochondrial dysfunction: formic acid participates in pyruvate and methane metabolism and is associated with the kynurenine pathway, while succinate, a key citric acid cycle intermediate, reflects impaired mitochondrial respiratory chain activity. Furthermore, PD and PSP patients shared elevated levels of carnitine and arginine, suggesting a common disruption in carnitine biosynthesis and mitochondrial fatty acid transport, as well as potential alterations in nitric oxide metabolism via arginine pathways. Conversely, MSA and PSP exhibited changes in taurine, a neuroprotective amino acid involved in osmoregulation and neurotransmission, and in the lipid species, pointing to shared lipid remodeling processes and tauopathy-associated membrane disturbances.

Interestingly, gut microbiota studies have indicated partially overlapping microbial shifts across PD and APS, yet with distinct taxa or metabolite patterns that may support early differentiation [89]. To the best of our knowledge, no metabolomics investigations have directly compared fecal profiles between PD and APS patients. Nevertheless, there is experimental evidence of syndrome-specific alterations in the fecal metabolome of PD patients compared to healthy controls, including reduced levels of branched-chain and aromatic amino acids [90] as well as changes in volatile organic compounds [91]. These findings suggest that fecal metabolomics could provide additional insights beyond those obtainable from other biofluids, capturing disease-specific signatures that may otherwise remain undetected.

Given the high sensitivity of metabolomics in detecting biochemical changes across multiple biofluids, integrating gut-derived metabolomics profiles of PD, PSP, and MSA patients with those from other matrices (rather than relying on a single biofluid) alongside clinical insight could not only contribute to highlight disease-specific biochemical alterations but also provide valuable diagnostic implications. Considering the multi-level complexity of these disorders, combining metabolomics with other omics approaches holds strong potential to improve early differential diagnosis, clarify disease-specific mechanisms, and guide the development of targeted therapeutic strategies.

## 7. Conclusions

This systematic review summarizes an up-to-date overview of recent advances in the application of metabolomics to Parkinson’s disease, focusing on studies published between 2019 and 2024. Beyond cataloguing findings, it provides a structured evaluation of study designs, analytical platforms, and statistical methods, even identifying recurrent methodological limitations that may considerably impact data comparability and the length of transition from bench to bedside.

A major contribution of this review lies in its comparative analysis of multiple biofluids (plasma, serum, CSF, saliva, sebum, and urine), offering a broader perspective on both central and peripheral metabolic alterations in PD. Despite the diversity in study protocols, several recurring metabolic signatures were observed, particularly involving amino acid metabolism, lipid dysregulation, mitochondrial dysfunction, oxidative stress, and microbial metabolites. These patterns suggest systemic metabolic derangements that may contribute to disease pathophysiology.

While variability in cohort size, confounder control, and duration of follow-up remains a limiting factor, the compiled evidence highlights the potential of metabolomics to support biomarker discovery and stratified medicine in PD. Particularly promising is the emerging role of peripheral biofluids such as saliva, sebum, and urine, which offer accessible alternatives to more invasive sampling methods like CSF, and a promising area for further research aimed at developing accessible diagnostic tools.

To advance this field and improve clinical translation, future research must prioritize longitudinal studies, robust validation strategies, and improved methodological standardization. In particular, the integration of metabolomics with other omics platforms, such as genomics, transcriptomics, and proteomics, holds transformative potential. These multi-layered approaches enable a more holistic understanding of the complex biochemical and molecular networks driving PD pathogenesis, and may uncover mechanistic links that remain invisible when each omics layer is studied in isolation. The multi-omic approach not only increases the robustness of biomarker discovery, by validating findings across independent biological layers, but also enhances mechanistic interpretation, providing insights into causal relationships and compensatory adaptations. Such integrated frameworks have the potential to identify subgroups of patients with distinct pathophysiological signatures, paving the way for precision medicine strategies in PD, from early diagnosis to targeted therapeutic interventions.

## Figures and Tables

**Figure 1 jcm-14-06277-f001:**
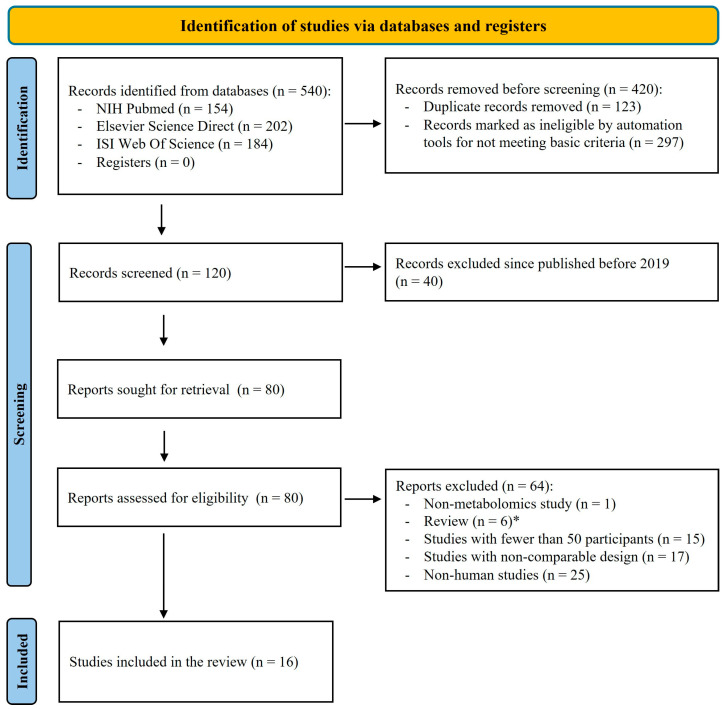
PRISMA 2020 flow diagram illustrating the study selection process for the present systematic review, based exclusively on searches of databases and registers. * These records initially indexed as original studies were later identified as reviews and excluded during manual screening.

**Table 1 jcm-14-06277-t001:** Summary table of selected article characteristics.

Authors’ and Patients’ Nationality	Study Design	Participants ^a^	Gender ^b^	Age (Years) ^c^	Sample Type	Experimental Platform ^d^	Statistical Analysis of Metabolomics Data ^e^
Glaab et al. (2019) [40] Germany	Integrative analysis	PD (60) Ct (15)	19 M/41 F (PD) 8 M/7 F (Ct)	65.7 ± 9.0 (PD) 65.1 ± 8.4 (Ct)	Plasma	GC-MS	Welch’s t test, SVM, RF, ROC analysis
Heilman et al. (2020) [41] USA	Case-control study	PD (97) Ct (90)	49 M/48 F (PD) 43 M/47 F (Ct)	66.9 ± 8.3 (PD) 66.6 ± 10.3 (Ct)	Plasma, CSF	HPLC	Mann–Whitney U-test, PLS-DA, OPLS-DA, ROC curve analysis
Kumari et al. (2020) [42] North India	Case-control study	PD (76) Ct (37)	14 M/23 F (PD) 56 M/17 F (Ct)	54.9 ± 7.8 (PD) 53.0 ± 8.6 (Ct)	Saliva	NMR	Mann–Whitney U-test, PLS-DA, OPLS-DA, ROC curve analysis
Kremer et al. (2021) [43] Germany, USA	Longitudinal analysis	PD (157) Ct (105)	106 M/51 F (PD) 70 M/35 F (Ct)	63.2 ± 9.6 (PD) 64.1 ± 8.9 (Ct)	CSF	LC–MS/MS	Linear regression, linear mixed-effect model, ROC curve analysis
Sinclair et al. (2021) [44] UK	Observational study	PD (218) Ct (56)	137 M/81 F (PD) 26 M/30 F (Ct)	70.0 ± 8.8 (PD) 54.3 ± 14.4 (Ct)	Sebum (from skin)	LC-MS	PLS-DA, ROC curve analysis, pathway enrichment analysis
Shao et al. (2021) [45] China	Case-control study	PD (223) Ct (169)	124 M/99 F (PD) 94 M/75 F (Ct)	66.3 ± 1.3 (PD) 6.2 ± 1.3 (Ct)	Plasma	LC-HRMS	Mann–Whitney U test, PCA, OPLS-DA, hierarchical cluster analysis, binary logistic regression analysis, ROC curve analysis
Mallet et al. (2022) [46] Italy, USA	Translational study	PD (129) Ct (53)	63 M/66 F (PD) 28 M/25 F (Ct)	61.4 ± 9.1 (PD) 62.4 ± 8.3 (Ct)	Serum	NMR	ANOVA, OPLS-DA, ROC curve analysis
Dahabiyeh et al. (2023) [47] Jordan	Case-control study	PD (50) Ct (45)	31 M/19 F (PD) 23 F/22 M (Ct)	64.2 ± 13.3 (PD) 59.4 ± 10.4 (Ct)	Serum	LC-MS/MS	Student’s *t*-test, ANOVA, PLS-DA
LeWitt et al. (2023) [48] USA	Case-control study	PD (50) Ct (50)	30 M/20 F (PD) 21 M/29 F (Ct)	69.5 ± 6.6 (PD) 66.0 ± 6.9 (Ct)	Serum, CSF	UHPLC-MS/MS	Student’s *t*-test, SVM, PLS, LASSO regression, pathway enrichment analysis
Paul et al.(2023) [49] USA	Metabolome-wide association study (MWAS)	PD (642) Ct (277)	406 M/236 F (PD) 129 M/154 F (Ct)	66.8 ± 10.4 (PD) 65.6 ± 12.8 (Ct)	Serum	LC-HRMS	Linear regression, logistic regression
Wang et al. (2023) [50] China	Case-control study	PD (104) Ct (111)	65 M/39 F (PD) 60 M/51 F (Ct)	59.4 ± 12.1 (PD) 57.2 ± 9.1 (Ct)	Urine	LC-MS	PLS-DA, RF, XGBoost, LASSO regression, ridge regression, ROC curve analysis
Dahabiyeh et al. (2024) [51] Jordan	Case-control study	PD (50) Ct (45)	31 M/19 F (PD) 22M/23 F (Ct)	64.2 ± 13.3 (PD) 59.4 ± 10.4 (Ct)	Serum	GC-TOF MS LC-MS/MS	ANOVA, PLS-DA, ChemRICH analysis
de Lope et al. (2024) [52] Luxembourg	Cohort-wide profiling	PD (549) Ct (590)	360 M/189 F (PD) 384 M/206 F (Ct)	66.0 ± 10.7 (PD) 61.7 ± 11.7 (Ct)	Plasma	LC-MS/MS	Linear regression, SVM, ROC curve analysis, pathway enrichment analysis
Hu et al. (2024) [53] UK	Prospective cohort study	PD (639) Ct (109.146)	409 M/235 F (PD) 50.308 M/58.838 F (Ct)	62.7 ± 5.6 (PD) 56.5 ± 8.1 (Ct)	Plasma	NMR	PCA, Cox proportional hazard model, logistic regression, ROC curve analysis
Hu et al. (2024) [54] China	Observational study	PD (75) Ct (31)	37 M/38 F (PD) 16 M/15 F (Ct)	66.1 ± 6.3 (PD) 64.1 ± 8.7 (Ct)	Plasma	LC-MS	PCA, OPLS-DA
Wang et al. (2024) [55] China	Case-control study	PD (99) Ct (91)	54 M/45 F (PD) 46 M/45 F (Ct)	60.9 ± 10.1 (PD) 61.7 ± 11.2 (Ct)	Plasma	LC-MS	Student’s *t*-test, Mann–Whitney U test, PLS-DA, KNN, RF, LASSO, SVM, ROC curve analysis

^a^ PD = Parkinson disease; Ct = controls; ^b^ Gender is reported as the number of males (M) and females (F) per group. ^c^ Age is reported as mean age ± standard deviation for both patient and control groups. ^d^ Abbreviations: GC-MS = Gas Chromatography-Mass Spectrometry; GC-TOF MS = Gas Chromatography-Time of Flight Mass Spectrometry; HPLC = High-Performance Liquid Chromatography; LC-HRMS = Liquid Chromatography-High Resolution Mass Spectrometry; LC-MS = Liquid Chromatography-Mass Spectrometry; LC–MS/MS = Liquid Chromatography-Tandem Mass Spectrometry; NMR = Nuclear Magnetic Resonance Spectroscopy; UHPLC-MS/MS = Ultrahigh-Performance Liquid Chromatography-Tandem Mass. ^e^ Abbreviation: ANOVA = Analysis of Variance; ChemRICH = Chemical Similarity Enrichment; LASSO = Least Absolute Shrinkage and Selection Operator; Linear SVM = Linear Support Vector Machine; OPLS-DA = Orthogonal Partial Least Squares Discriminant Analysis; PCA = Principal Component Analysis; PLS = Partial Least Squares; PLS-DA = Partial Least Squares Discriminant Analysis; RF = Random Forest; RM ANOVA = Repeated Measures Analysis of Variance; ROC = Receiver Operating Characteristic; SVM = Support Vector Machine; XGBoost = Extreme Gradient Boosting.

**Table 2 jcm-14-06277-t002:** Quality assessment of included studies based on NOS ^a^.

Study	Selection	Comparability	Outcome	Total
Glaab et al. 2019 [40]	●●●●	●●	●●●	9/9
Heilman et al. 2020 [41]	●●●	●	●●	6/9
Kumari et al. 2020 [42]	●●●	●	●●	6/9
Kremer et al. 2021 [43]	●●●	●●	●●●	8/9
Sinclair et al. 2021 [44]	●●●	●●	●●●	8/9
Shao et al. 2021 [45]	●●●●	●●	●●●	9/9
Mallet et al. 2022 [46]	●●●	●●	●●	7/9
Dahabiyeh et al. 2023 [47]	●●●●	●	●●●	8/9
LeWitt et al. 2023 [48]	●●●●	●●	●●●	9/9
Paul et al. 2023 [49]	●●●●	●	●●●	8/9
Wang et al. 2023 [50]	●●●	●●	●●	7/9
Dahabiyeh et al. 2024 [51]	●●●●	●	●●●	8/9
de Lope et al. 2024 [52]	●●●●	●●	●●●	9/9
Hu et al. 2024 [53]	●●●●	●●	●●●	9/9
Hu et al. 2024 [54]	●●●	●●	●●●	8/9
Wang et al. 2024 [55]	●●●	●	●●	6/9

^a^ Assessment is based on three domains: selection (max 4 points), comparability (max 2 points), and outcome (max 3 points). Total score out of 9.

**Table 3 jcm-14-06277-t003:** Summary of the major metabolic alterations, associated biofluids, and key metabolites reported in PD.

Metabolic Disregulation	Metabolites ^a^	Biofluids
Amino acid metabolism	*N*-acetylglutamate (NAG) ↑, alanine ↑, aspartate ↓, g-aminobutyric acid (GABA) ↑, glycine ↑, glutamate ↑, histidine ↑, isoleucine ↑, phenylacetyl-L-glutamine ↑, phenylalanine ↑, pyroglutamic acid ↑, serine ↑, threonine ↑, tryptophan ↓, tyrosine ↑, valine ↑	plasma [41,53], serum [46,49], saliva [42]
Energy metabolism	acetoacetate ↑, acetoin ↑, alanine ↑, citric acid ↑, 3,3-dimethylglutaric acid ↓, β-hydroxybutyrate ↑, orotic acid ↑, oxoglutaric acid ↑, pantothenic acid ↓, pyruvate ↑	plasma [55], saliva [42], serum [46,49], urine [50]
Gut microbiota-derived metabolites	acetate ↑, butyrate ↑, p-cresol ↑, p-cresol glucuronide ↑, p-cresol sulfate ↑, phenylacetyl-L-glutamine ↑, propionate ↑, TMAO ↑	plasma [45,52,53], saliva [42], serum [49]
Lipid metabolism	- acylcarnitine (e.g., benzoylcarnitine ↑) - free fatty acids (e.g., hexadecanoic acid ↑, dodecanoic acid ↑) - phospholipids (comprising glycerophosphocholines, lysophosphatidylcholines, phosphatidylserines, 1-arachidonoyl-phosphatidylinositol) - sphingolipids (such as ceramides) - short-chain fatty acid (e.g., butyrate ↑) - eicosanoids, such as hydroxyeicosatetraenoic acid (12-HETE) ↑	plasma [40,45,53,54,55], sebum [44], serum [47,49,51]
Neurotransmitter metabolism	Tryptophan–kynurenine pathway: *N*-acetylglutamate (NAG) ↑, g-aminobutyric acid (GABA) ↑, glycine ↑, 3-hydroxyanthranilic acid ↓, 3-hydroxykynurenine ↑, indolelactic acid ↓, kynurenic acid ↓, kynurenine ↓, quinolinic acid ↑ Dopamine metabolism: 3-methoxytyramine ↑, N-acetyl-l-tyrosine ↑, 3,4-dihydroxyphenylacetic acid (DOPAC) ↓, homovanillic acid (HVA) ↓	CSF [41], plasma [41,45], saliva [42], serum [49] CSF [43], urine [50]
Oxidative stress	threonic acid ↑, uric acid ↑, vanillic acid ↑, xanthine and derivatives ↑	plasma [40,52], serum [51], urine [50]
Polyamine metabolism	*N*-acetylputrescine ↑, *N*-acetylcadaverine ↑, *N*8-acetylspermidine ↑, L-ornithine ↑	CSF [48], serum [51]

^a^ Arrows (↑/↓) indicate the reported direction of change in PD compared with controls. For lipids, the molecular class rather than individual species is reported, as, in many cases, different species within the same class showed inconsistent trends depending on sex [45]. When sex-specific differences were observed, no arrow is provided.

## Data Availability

Not applicable.

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
