# Peer review of "Parkinson’s Disease Through the Lens of Metabolomics: A Targeted Systematic Review on Human Studies (2019–2024)"

_jcm, 2025, doi:10.3390/jcm14176277_

Round 1

Reviewer 1 Report

Comments and Suggestions for Authors

The analysis of Parkinson's disease (PD) remains a timely and evolving issue, I have the following comments regarding this review:

  • the aspect of PD pathophysiology could be enriched by the contemporary hypotheses with regard to extensive analysis of neuroinflammation
  • the analysis would be interesting if acknowledging the possible significance of PD subtypes in the metabolomic evaluation
  • the metabolomic could be additionally analyzed in the context of possible significance of the analysis in the examination of entities overlapping with PD as Progressive Supranuclear Palsy (Ref. Progressive Supranuclear Palsy-Parkinsonism Predominant (PSP-P)-A Clinical Challenge at the Boundaries of PSP and Parkinson's Disease (PD). Front Neurol. 2020;11:180. Published 2020 Mar 10. doi:10.3389/fneur.2020.00180)

Reviewer 2 Report

Comments and Suggestions for Authors

The review by Cannas and colleagues describe an investigation of human metabolomics in Parkinson’s disease with reference to studies published 2019-2024. A comprehensive scheme was utilized to filter out and subsequently include 16 original studies. The review is rigorous and follows PRISMA 2020 guidelines. Different aspects of the studies reviewed were highlighted including the tissues used, platform applied, and the interpretation of results. The study was well designed and organized, the reviewed data represent the most recent and robust data available. The data are thoroughly reviewed although validation of the results through data mining functional information would helpful.  Some suggestions for improving the text can be found below.

  1. The rationale for including studies only from 2019-2024 was not provided.
  2. As mentioned by the authors, several other metabolomics reviews have been published recently. While the authors describe the content of those previous reviews, the additional value (novelty, focus, quality assurance etc.) from the current review should be more explicitly stated.
  3. How does the human PD metabolomics data compare to those in animal models ? Do the human results reviewed here recapitulate the results from experimental animals ?
  4. Given the common use of CSF for biomarker development, the section on its application is shallow. More depth should be provided on CSF biomarkers.
  5. The data sets reviewed included mostly older adults. Given the long prodromal stage of Parkinson’s disease, the authors should comment on whether the results may be different depending upon the sampling period during the course of the disease.

Reviewer 3 Report

Comments and Suggestions for Authors

The manuscript titled “Parkinson’s Disease Through the Lens of Metabolomics: A Targeted Overview on Human Studies (2019–2024)” presents a well-organized and timely systematic review of human metabolomic studies in Parkinson’s disease (PD). 

While the review is interesting and informative, there are some major concerns:

1, While the review is systematic, there is no meta-analysis or pooled effect estimation. Quantitative synthesis (e.g., odds ratios or AUCs) would strengthen conclusions about biomarker robustness. The authors should include a table or forest plot of key biomarkers across studies. This would help visualize consistency and clinical relevance of candidate biomarkers across biofluids and platforms.

2, Although confounder adjustment is mentioned, few details are given on how variables like age, medication (especially levodopa), comorbidities, and diet were controlled across studies. This weakens causal interpretation. Furthermore, analytical heterogeneity is mentioned, but more elaboration is needed on proposed standard operating procedures (SOPs) or guidelines for sample collection, processing, and data normalization. While sex is reported, its influence on metabolic profiles is not analyzed or discussed—yet sex-specific metabolism is an emerging area in PD.

3, The review acknowledges medication effects on metabolite levels but does not sufficiently detail how this was addressed or quantified across studies. For example, levodopa dramatically alters tyrosine and dopamine metabolism.

4, Most studies compare PD patients vs. controls, but PD is a progressive disorder. The review would benefit from stratifying or discussing results by disease stage or duration, which may affect metabolite levels (e.g., ketone bodies, BCAAs). The conclusion briefly suggests multi-omics, but specific examples (e.g., metabolomic-genomic integration for GBA or LRRK2) would clarify translational potential.

5, The discussion does not sufficiently address the limitations and variability of individual biofluids (e.g., saliva’s susceptibility to contamination or diet, sebum's sampling standardization challenges).

6, While multiple statistical methods are listed (PCA, PLS-DA, ROC, etc.), their comparative strengths and weaknesses or appropriateness for metabolomic data are not critiqued.

Round 2

Reviewer 1 Report

Comments and Suggestions for Authors

The perspective on boundaries and overlaps with related disorders e.g. PSP-P was not sufficiently outlined.

Reviewer 3 Report

Comments and Suggestions for Authors

The authors have addressed all concerns properly. No further concerns. I suggest accepting it in the current form

Author Response

We sincerely thank the reviewer for the positive evaluation of our revised manuscript and for recommending it for acceptance in its current form.

Round 3

Reviewer 1 Report

Comments and Suggestions for Authors

In my opinion the current version of the manuscript was sufficiently revised. As a minor comment, I would suggest to additionally provide a more extensive overview on metabolomic perspective of related disorders (e.g. other parkinsonisms apart from PSP-P).
